# Anti-Seismic Performance Evaluation of Waterproofing Materials for Underground Pile Wall Structures

**DOI:** 10.3390/ma14195719

**Published:** 2021-09-30

**Authors:** Seung-Jin Lee, Soo-Yeon Kim, Sang-Keun Oh

**Affiliations:** 1Graduate School Department of Architecture, Seoul National University of Science and Technology, Seoul 01811, Korea; mulgae22@hotmail.com; 2Construction Technology Research Institute, Seoul National University of Science and Technology, Seoul 01811, Korea; ksr1115@seoultech.ac.kr; 3School of Civil Engineering Architecture, Environment of Hubei University of Technology, No.28, Nanli Road, Hong-shan District, Wuchang, Wuhan 430068, China

**Keywords:** pile walls, waterproofing, anti-seismic performance, evaluation method

## Abstract

This study introduces and demonstrates the application of an experimental regime for anti-seismic performance evaluation of waterproofing materials used for concrete pile walls. Concrete pile walls are subject to high degrees of seismic load, and the resultant stress can affect the waterproofing integrity of the structure, but there is currently no existing methodology or standard for evaluating this property of waterproofing materials. To propose and conduct this evaluation, a new testing apparatus was designed and manufactured to test an installed waterproofing material’s seismic resistance performance. Under three different inclined angle conditions (0°, 10°, 20°), each with three different rotation speed conditions (10, 20 and 30 rotations per minute), three types of waterproofing materials were subjected to 30 s of increasing seismic stress and tested for their waterproofing performance. Waterproofing performance was determined by whether the specimen installed with the respective type of material was able to prevent leakage path formation during the seismic stress, and the performance was summarized and compared based on the average results for four specimens of each material type and the duration before leakage occurrence. Results of the demonstration testing yielded significantly different results for the tested material types, prompting the need to further investigate different types of waterproofing materials, products, and techniques for a comprehensive understanding of waterproofing material response properties against seismic stress. The demonstration process shown in this research was intended to serve as a proposal for the development of these performance evaluation criteria, methodologies, and equipment for possible future application.

## 1. Introduction

In the construction of buildings and civil engineering projects, pile wall construction is a commonly employed method for setting the foundation of the structure. In highly congested city areas where new tunnels, underground railways, or common ducts are being constructed, it is difficult to secure a large construction site whereby pile walls can be used. The primary purpose of pile walls is for ground retention, especially for construction of large-scale infrastructure in poor geographical environments [1]. Construction of pile walls comprises forming a continuous wall structure by drilling piles one after another in a secant structure. While the pile walls are designed to be impervious, they are not waterproof in nature due to the formation of numerous joints between the piles, which can become sources of leakage [2]. Therefore, when waterproofing pile walls, higher quality waterproofing materials, construction methods, and quality control standards are required compared to general waterproofing construction. This is because a high degree of adhesion durability is required in areas where severe bending is anticipated due to the presence of foreign substances (soil, groundwater, etc.) affecting the integrity of the pile wall. In addition, during the design of waterproofing, the ability to respond to continuous vibrations generated due to local sources of load should be considered [3]. Moreover, the ability to respond to the behavior of the structure itself according to the contraction and expansion (freeze-thaw effect, for example) that occurs due to various causes in all concrete structures must also be secured. 

Furthermore, although it occurs sporadically and intermittently, the seismic response ability to earthquakes with very large wavelengths must also be verified to determine the applicability of the waterproofing of the composite wall under the worst conditions. Given conventional circumstances where waterproofing materials are verified only by their physical property assessment, it is difficult to verify their corresponding mechanical properties [4]. Due to these difficulties, it is currently impossible to verify the quality of the waterproofing for pile wall structures. Therefore, in this study, a new seismic performance testing apparatus was developed with the purpose of evaluating the mechanical and waterproofing performance of waterproofing materials. The main focus and proposed novelty for this study was to develop an evaluation method and promote secure application of waterproofing materials for pile walls. Due to the nature of secant pile construction, conventional installation methods may not suffice to achieve secure installation as various types of angles, corners and overlaps are present. In such cases, waterproofing materials that are commonly known to be able to achieve high and stable adhesion on regular surfaces may be susceptible to adhesion failure from just minor vibration effects (such as earthquakes) due to the initial lack of proper installation, as it commonly is difficult to secure proper workmanship on vertical angular column type walls. In this regard, it is deemed necessary to take into consideration the effects of vibration, movement behavior of concrete, and earthquake response performance; hence, new evaluation criteria were also proposed and the corresponding evaluation regime was implemented on three different waterproofing materials as a demonstration for determining the applicability of this new test method. 

## 2. Theoretical Discussion

### Investigation of Existing Domestic and Foreign Quality Control Standards Applicable to Wall Waterproofing

Currently, in Korea where this research was conducted, Korean Construction Standard (KCS) 41 40 13 (“Underground exterior waterproofing construction”) and KCS 41 40 04 (“Adhesive flexible sheet waterproofing construction”) [5,6] are mentioned as quality standards applicable to underground wall waterproofing in Korea. For material quality standards, KS F 4935 (“Adhesive flexible rubber asphalt-based injection sealing material for water leakage repair”) and KS F 4911 (“Synthetic polymer based waterproofing sheet”, and “High adhesive composite waterproofing sheet”) are currently in the process of enactment [7,8,9]. 

Internationally, many different quality standards are used in the field, but are not specific to pile wall waterproofing construction. BS EN 8102 (“Code of practice for the protection of below ground structures against water from the ground”) provides some guidelines on the importance of waterproofing with retaining walls [10], and companies in the U.S. commonly adhere to ASTM D 7832 (“Standard Guide for Performance Attributes of Waterproofing Membranes Applied to Below-Grade Walls/Vertical Surfaces (Enclosing Interior Spaces)”) [11] performance requirement specification for waterproofing materials when pile walls need to be waterproofed, but waterproofing of the secant pile wall is not a clearly defined category of standards and design guidelines in the United States. In Japan, it was also found that waterproofing of walls is not implemented on a standardized basis. In the case of underground construction using existing joint wall construction methods, it was found that waterproofing of pile walls is not necessarily mentioned or compulsorily implemented. However, it was confirmed that papers on methods of waterproofing pile walls have been published by Japan-related associations and research institutes [12,13]. 

Based on a literature review and research, it is evidenced that pile wall waterproofing is practiced in most countries, but standards or details on performance requirements have yet to be developed. A common point in existing international quality standards for waterproofing materials, including the investigated standards, is that they are standardized for the purpose of verifying only basic physical properties centered on materials. That is, they are applied as a criterion for judging the properties of the waterproofing materials themselves; thus, it is often difficult to predict their actual performance in the field or to judge the responsiveness to the degradation environment of the part where the waterproofing materials or construction methods have been applied.

## 3. Underground Wall Waterproofing Construction and Response Environment Analysis

### 3.1. Underground Wall Waterproofing Construction

Pile wall construction is required for the exterior wall on the opposite wall of a building in a site situation where sufficient excavation work is difficult [14]. Concrete wall waterproofing refers to waterproofing work carried out for the purpose of preventing water leakage from the groundwater environment and protecting the structure in the constructed earth wall or pile, etc. [15]. In general, a pile wall structure that includes waterproofing is installed in the following order: ground treatment > waterproofing layer construction > waterproofing layer fixation (reinforcement around fixed hardware) > joint reinforcement > rebar reinforcement > concrete pouring [16]. A typical example of pile wall construction is shown in Figure 1 below.

Pile wall waterproofing construction is often conducted under disadvantageous environmental conditions (examples shown in Figure 2 below) that include high humidity and the presence of foreign substances and laitance. As the vertical wall surfaces of pile walls are rounded with multiple joints, high-quality and durable waterproofing materials are difficult to apply and require high-quality workmanship. In particular, the continuous joint movement of the pile walls can cause waterproofing materials to deform, crack, peel off or form various types of defects [17]. 

The premise of this study and proposal for a new evaluation method was centered in assessing the quality of waterproofing materials required for wall waterproofing in terms of their response performance to vibrations, movement behavior, and earthquakes. As such, the parameters that affect the stability of waterproofing material performance were outlined first in order to derive logical evaluation criteria.

### 3.2. Environmental Degradation Effects on Underground Structures That Affect Waterproofing Materials

As mentioned above, waterproofing works to be constructed in an underground pile wall section must respond to various natural, physical, and artificial conditions under the geographical environment of the basement.

#### 3.2.1. Natural Environment

The natural environment includes snow, rain, hail, wind, typhoons, and air temperature transmitted from the ground. In addition, there is dew condensation that occurs both above ground and underground, and salt damage that affects building structures located in coastal areas. Refer to Figure 3 for an illustration of overall effects of natural environment on structures.

#### 3.2.2. Mechanical Effects

Intermittent but regularly occurring vibrations caused by vehicles, subways, etc., and contraction and expansion behaviors (tensile or shear stress) caused by behavioral movement can affect the concrete substrate as illustrated in Figure 4. In addition, there are other factors such as water pressure, flow velocity, wet environment, soil pressure, floating settlement, vibration, and earthquakes.

### 3.3. Basic Theoretical Analysis of Waterproof Performance Design in Response to Vibration, Behavior, and Earthquake

Shock waves from sources such as vibrations, behaviors, and earthquakes that affect a concrete structure are aperiodic and are transmitted in the form of a very complex wavelength. In addition, the shape, scale, direction, etc. of each wave is different based on the Richter scale [18]. Quantitatively calculating these wavelengths is a very difficult task even in the field of seismic engineering, and even more so as far as determining how the varying degrees of seismic wave may affect the waterproofing materials. Seismic waves are classified into P-waves (primary), S-waves (secondary), L-waves (love), and R-waves (Rayleigh) as shown in Table 1 below. The wave shape, particle motion direction, velocity, and characteristics of each seismic wave are as follows.

When represented in the form of a beam element, the above types of seismic waves can be depicted as the images shown in Table 2 below. The different types and forms of wavelengths affect the beam element in different ways.

### 3.4. Seismic Design of Concrete Building Structures

#### 3.4.1. Seismic Performance and Demand Response Spectrum of the Structure

In general, the ability to respond to seismic waves is called seismic performance for concrete structures [19]. When an earthquake occurs, the elastic energy emitted around the point (seismic circle) is transmitted in the form of a seismic wave with force (load). Seismic performance refers to the ability to withstand an earthquake well in response [20]. Wavelengths for seismic response evaluation of construction structures are applied based on the frequency (or period) through seismic analysis. The frequency of the wave is reflected by the demand response spectrum, which defines the earthquake motion for a building structure earthquake proofing [21]. The demand response spectrum refers to the value of the response period or frequency (displacement, speed, acceleration, etc.) generated in the structure by the external load, and is used to define the earthquake motion to verify the seismic performance [22]. The response spectrum is mainly used in nuclear power plants and reflected in the earthquake-resistant design. Figure 5 shows the basic concept of the response spectrum, and the graph shows the response by maximum response time according to the response period.

As such, the seismic design of the structure defines and reflects the required spectrum according to the response type corresponding to the form of wavelength, but in the case of the waterproofing layer of the composite wall being reviewed in this study, the thickness is approximately 3 mm, whereby the effect on structural performance is negligible. As certain types of waterproofing materials do not have resonant frequency and are considered as non-structural components in concrete structures, it is difficult to claim that the wavelength frequency and response spectrum are applicable to waterproofing materials for seismic design. Waterproofing materials, however, should ideally maintain a near homogenous adhesive bonding to the concrete substrate surface in order to make the structure/member impervious to water penetration, and where the concrete element is affected by the seismic load, the quality of waterproofing material will naturally be affected as well [23]. Therefore, it is judged that it is necessary to define the demand response spectrum of the waterproof layer with regards to their waterproofing performance by a new concept for the analysis of the impact on the waterproof layer reflected in the seismic performance design of the structure, mainly by investigating the affected adhesion performance and waterproofing capacity.

#### 3.4.2. Seismic Design of Non-Structural Elements and Response Spectrum

The seismic design of non-structural elements is intended to minimize damage to a structure caused during the event of an earthquake, as well as damage to building materials constructed with interior and exterior materials, such as degradation due to the waves of the earthquake, and refers to design standards that reflect building anti-seismic performance. In 2018, Korea divided non-structural elements of buildings into architectural, mechanical, and electrical, and prepared seismic design standards. Building non-structural elements (members) are classified into 13 types, including internal non-structural walls and partition walls, cantilever members, external non-structural wall members and joints, among the earthquake-resistant design standards of KDS 41 17 00 (2019). Among them, in the case of SPS-F KOCED 0007-7419, which was established as a group standard in 2021: the vibration table test method for seismic performance evaluation of suspended ceilings, where the setting is similar to the demand response spectrum transmitted to the waterproof layer installed onto the concrete structure. 

Furthermore, this standard stipulates the vibration table test method to evaluate the seismic performance of a suspended ceiling composed of lightweight steel frame, ceiling finishing material, molding, etc. and installed floating on a structure or other non-structural structure. In particular, the demand response spectrum used in this test method uses the floor response spectrum presented in ICC-ES AC 156 for the floor where the suspended ceiling is installed. This is evaluated in the form of generating spatial coordinates by time by calculating the three seismic waves by scaling the seismic waves along the X, Y, and Z axes, performing numerical analysis, and then calculating the displacement response for each axis. That is, the acceleration time history is prepared and presented so that the demand response spectrum contains energy components from 1.3 Hz to 3.3 Hz. Refer to Figure 6 below for details.

The United States employs a seismic design for non-structural components in compliance with the American Society for Testing and Materials (ASTM) E 2026 Standard Guide for the Assessment of Earthquake Risks in Buildings, where it states that non-structural components are those of a building system subject to vertical or lateral loads and are not considered as a variable for defining seismic resistance. However, according to a case reviewed separately in “Development of improved seismic design and innovative control approaches of non-structural components to improve seismic performance of buildings and non-structural components (NSCs)”, seismic performance failure can be attributed to either direct or indirect consequences of NSC damage during earthquakes, and it follows that a need for seismic design for non-structural elements is mentioned. In addition, in 2015, the NEHRP standard in the US was revised to ASCE 7 to include an item for the minimum design load for non-structural components. This section specifies that the seismic design factor should be calculated considering the anchorage, deflection, and displacement factors of non-structural components such as elevators, suspended ceilings, electrical components and tracks of structures. As such, waterproofing materials can be evaluated for their waterproofing performance in response to seismic stress. 

### 3.5. Reinterpretation of Demand Response Spectrum on Waterproofing Material

In the seismic design of structural and non-structural elements, it was confirmed that the natural frequency, acceleration response, and displacement response of the response spectrum required for each building member were different. Based on these matters, it is judged that the demand response spectrum to respond to vibrations, behaviors, and earthquakes should be newly defined and reinterpreted under the condition that the waterproof layer is directly or indirectly attached to the structure. That is, as seismic design standards required for each structure, structural member, and non-structural element are set differently, it is necessary to reset the demand response spectrum for vibration and behavior transmitted to the waterproofing layer and seismic waves. Therefore, in this study, the factors affecting the design seismic wave, natural frequency, acceleration response, and displacement response were reinterpreted as the vibration, behavior, and demand response spectrum required for the waterproofing layer and can be defined based on the following parameters.

#### 3.5.1. The Type of Seismic Wave That Affects the Waterproofing Material

As investigated above, the design seismic wave is transmitted to the structure in the form of various wavelengths such as compression wave, longitudinal wave, shear wave, surface wave, and long wave. When these wavelengths are cut in the structure, they are cut in various angles and directions according to the shape of the wavelength. The order in which the wavelength is transmitted is first transmitted as a direct shock wave from the structure, and then directly or indirectly transmitted to the waterproofing inset attached to the structure. At this time, it is predicted that the seismic wave transmitted to the waterproofing layer will be transmitted at a slightly reduced angle and direction than the direct impact wave received by the structure. If this is interpreted as a design seismic wave that affects the waterproofing layer, it can be defined as a beat shape with various angles and rotations. In other words, it is defined as a form of periodic counting and weakening of wavelengths of different frequencies by causing interference from the surroundings. This can be interpreted as the wave that the waterproof layer must respond to vibration, behavior, and earthquake effects are transmitted through the angle and rotation in the form of a beat. Therefore, based on the analysis of vibration, behavior, and seismic waves affecting the waterproofing layer, a test device that can be reflected in standard angles and rotations should be designed.

#### 3.5.2. Natural Frequency of Waterproof Material

In this study, the design form of seismic waves affecting the waterproofing layer was defined via varying angles and rotations. Reflecting seismic waves and analyzing the natural frequency, acceleration, and displacement response of the waterproofing layer, the natural frequency can be defined by angle, rotation, and speed. As mentioned above, the frequency is interpreted as a slightly reduced shaking and movement as the direct shock wave delivered to the structure is transmitted to the waterproofing layer. It should be designed with a standard angle, rotation, and speed in a way that can verify the shaking and movement that occurs at this time. That is, it is required to design a verification device for the natural frequency to which the waterproofing layer must respond under various angles, rotations, and speeds.

#### 3.5.3. Acceleration Response of Waterproof Material

Even in the response of acceleration, it is possible to analyze the response by verifying the natural frequency within a certain range by controlling the speed according to time. In other words, it is judged that a device capable of controlling the standard speed for the waterproofing layer and programmable response is needed.

#### 3.5.4. Displacement Response of Waterproof Material

It is judged that the displacement of the waterproofing layer can be defined as rotation. The waterproofing layer should be designed so that the displacement through rotation about a certain axis can be confirmed. In other words, it is predicted that vibration, behavior, and earthquake displacement can be verified through displacement analysis of the waterproofing layer by controlling the rotation of the waterproofing layer around a certain axis and reproducing the response in the program.

As part of the high-durability quality verification for the waterproofing layer reviewed above, the behavior, vibration, and response to earthquakes were reinterpreted according to the design seismic wave, natural frequency, acceleration response, and displacement response. Through this, we defined rotation, speed, and angle, and design a verifiable test device based on the defined content to evaluate its effectiveness.

## 4. Experimental Regime

In this study, among the factors affecting the waterproofing layer, the required response spectrum of vibration, behavior, and earthquake was reinterpreted and set as rotation, speed, and angle. Based on the set contents, the design of the quality verification test equipment for the influencing factors was carried out. The design of the verification device is as follows.

### 4.1. Device Basic Configuration

The basic configuration of the quality verification test device was designed by partitioning the equipment into components, mainly compartmentalized into control panel, motor device for simulating seismic stress, and leakage checking device. The detailed design of each test device is as follows.

#### 4.1.1. Experimental Equipment Introduction

The equipment is designed to be able to control all mechanical operation that can simulate the vibration and behavioral movement of concrete subject to seismic stress. The main components were manufactured with the basic functions including control monitor, device control button, power switch, power supply, emergency stop button, etc. The control monitor and the power switch enable the tester to control the response spectrum based on the rotation speed and angle required for testing the waterproofing material. In addition, the equipment consists of operation, power, and stop buttons for driving and operation. Refer to Figure 7 for an illustration of the experimental equipment.

#### 4.1.2. Seismic Simulator Device Explanation

The seismic simulator device is designed to allow the application of rotation (direction), speed, and angle at which the specimen installed with waterproofing material can be subjected to a simulated load parameter similar to seismic load during earthquakes in concrete structures. In particular, the device is designed to operate and simulate up to three-dimensional stress by controlling the angle of three-axis (X, Y, Z) direction, rather than the conventional horizontal or vertical physical stress generation, so that various wavelengths of vibration, behavior, and seismic waves can be simulated. The main components are (1) a drive motor, (2) power transmission shaft, (3) angle control drive unit, and (4) a rotation device. Refer to Figure 8 below for an illustration of this device component. 

#### 4.1.3. Leakage Checking Process (Outlet)

The leakage checking method was designed with the intent to assess the tested waterproofing material’s ability to maintain its waterproofing performance throughout the seismic simulation. The compartments of the testing equipment allow fresh water to be placed above the waterproofing material specimen, with a leakage outlet situated at the bottom of the equipment. A sensor device will immediately alert the tester if leakage occurs due to the seismic simulation affecting the integrity of the waterproofing material. 

### 4.2. Stress Generation Principle and Range Setting

#### 4.2.1. Stress Generation Principle and Method Using Motor Device

The motor device within the equipment is designed to be operated according to the setting of rotation, speed, and angle based on the basic principle of triaxial stress generation in the form that affects the waterproof layer in the vibration and behavior transmitted to the building structure and the wavelength of the earthquake. Stress can be generated in the three dimensions of the vertical (y) axis, the left and right (x) axis, and the shear (z) axis to enable the generation of three-dimensional stress. Relative to the relevant and applicable types of stress for the waterproofing materials, simple categories were derived for the purposes of demonstration: tensile stress (when panels attached to the motor device are anchored at an obtuse angle format), compressive stress (when panels attached to the motor device are anchored at a reflex angle format), and shear stress (when panels attached to the motor device are anchored at an angle format, parallel to one another). A schematic diagram of the concept is shown in Figure 9 below.

In addition, a total of four separate 3-axis stress generating devices were designed, able to rotate in four directions, and each manufactured operating device was arranged in a positive, front, back, left, and right direction to provide their respective loading force. Each device operates such that the rotation, speed, and angle of the seismic simulation can be controlled separately and be applied. This is intended to simulate an environment where a continuous waterproofing layer can be installed on panels (1 × 1 m^2^), and various types of waterproofing materials (such as sheet materials and sheet coating composite waterproofing materials) can be tested. For the demonstration testing, waterproofing materials were prepared in rolls directly from factory manufacturing. For future application, evaluation is possible by reproducing even the two-layer overlap joint, three-layer overlap joint, and four-layer overlap joint, which are one of the weak points of waterproofing. For this testing, for consistency and demonstration purposes, waterproofing materials installed for specimen preparation were applied with 3 mm thickness to accord with the thickness of the sheet type material. Refer to Figure 10 below for an illustration and concept of the specimen for the proposed testing method.

#### 4.2.2. Stress Generation Range Setting Capacity Using Motor Devices

Rotational stress (seismic simulation) can be transmitted from all directions on the left and right when the wave transmitted from vibration, behavior, and earthquake reaches the waterproofing layer, so it is designed to drive both left and right rotations. The speed for the vibration table test method outlined seismic performance evaluation of suspended ceilings established as a Korean group standard in 2021 (SPS-F KOCED 0007-7419) was referred to for establishing the experimental conditions. To reflect the range of 1.3 Hz to 33.3 Hz frequency capacity of common seismic load, the equipment was calibrated to be operable at range of 1 to 30 RPM (maximum revolutions per minute). Angle control was designed to be operable in the range of 0 to 20° to quantitatively implement the wave shape of P wave, S wave, L wave, and R wave according to the size of the vibration and behavior transmitted to the waterproofing layer and the wavelength of the earthquake. Refer to Figure 11 and Figure 12 below for details on the stress generation (rotation) device.

The speed controller was separately mounted on each of the four actuators to allow each stress run. In terms of angle, it can be operated at an angle ranging from 0 up to 20° for each of the 4 motor device units by installing it individually at the lower part of the driving unit. That is, all four motor devices are designed so that speed, angle, and rotation can be adjusted individually. In addition, control of speed, angle, and rotational stress was made to be independently controllable in the control system. Refer to Figure 13 below for details and illustration.

### 4.3. Experimental Demonstration Process and Results

#### 4.3.1. Waterproofing Material Specification for Specimen Preparation

For this demonstration, three types of waterproofing materials were selected for testing, all with distinctly different characteristics and physical properties to highlight the applicability and potential use of this experimental regime and equipment: siliceous powder-type coating waterproofing material for cement-based waterproofing (CM), urethane-based coating material (LM) for membrane waterproofing, and self-adhesive waterproofing sheet (SM). Based on the international setting research discussed in the previous sections, product types with the most frequent usage in recent construction history were surveyed. For the CM and LM type, manufacturer specification was employed for installation, maintaining a thickness of 3 mm across the 1 × 1 m^2^ panel surface area. For the SM type, four separate sheets (550 × 500 mm^2^) were used, to be applied on the panel surface area, with overlap of up 50 mm forming at the joints of the panels. Among the three types, materials with the clearest variances were chosen intentionally, as the demonstration of the test method is intended to illustrate the difference in the performance of the different classification of the materials. Refer to Figure 14 for details of the material specifications.

#### 4.3.2. Experimental Conditions 

For the respective waterproofing membrane types, three angle conditions (ranging from 0 to 20°, increasing at an interval of 10) were applied, whereby a sub-condition of rotation speed ranging from 10 to 30 (intervals of 10) RPM were applied. For 10° angle condition, the layout of the motor device was set to tensile stress simulation condition (refer to Figure 9) condition, and for 20° condition, the motor device layout was set to shear stress simulation condition (refer to Figure 9). For each sub-condition, four separate specimens were prepared for each waterproofing membrane type, whereby seismic stress was simulated for 30 s. During testing, the ambient conditions were set to room temperature (23 ± 1 °C), and relative humidity (RH) of 60 ± 5% such that materials before and during testing would not undergo drastic property changes due to temperature or humidity settings.

During testing, specimens that did not produce any leakage for the duration of the 30 s were considered to have passed (marked with “O” in Table 3 below), while for specimens that produced leakage, the leakage time from the start of testing was recorded accordingly (marked with “X”). The average results were determined first by whether the minimum of up to three out of four specimens passed the testing or not, and the average time was determined as well. 

#### 4.3.3. Experimental Results

The demonstration experiment showed that the materials had distinctly different results correlative to the waterproofing materials and their characteristics. The CM type response was shown to be the lowest, as was expected of a brittle type material. Most of the specimens were not able to withstand one rotation from the testing device once testing entered the angled condition stages. Mode of failure was consistently fracturing at the joints between the panels. For the LM and SM types, the results were more varying. The performance of the LM type consistently decreased in terms of the duration of seismic stress application, and with the exception of the first condition (0°, 10 RPM), the LM type materials were not able to pass the testing throughout the rest of the conditioning. This is also an indicator of the material characteristics, and the results showed that while the LM type was able to display resistance against seismic stress under the most default conditioning, as soon as the rotation speed and angles were applied, the performance gradually decreased (although some specimens were able to pass the testing under higher stress conditioning). The SM type, however, showed more extreme variation in its results, which on average seemed to display passing performance (up to 10°, 30 RPM), in cases where failure did occur, it occurred almost as soon as the testing started. Mode of failure for the LM type in all cases involved material fracturing at the joints between the panels, in the same way as CM, but for SM, material fracturing or any other forms of cohesive failure were not apparent. Instead, failure was found mostly at the sheet overlap sections, indicative of adhesion failure. The consistent results up to the 10°, 30 RPM conditioning showed that, comparatively, the SM type had the highest anti-seismic response performance among the three material types, but upon investigating the failure mode, it was also apparent that the workability for securing high-quality performance was more difficult due to the nature of the sheet material having to form overlap sections during installation. Results of the demonstration experiment of the three types of waterproofing materials are outlined in Table 3 below and summarized in Figure 15 below. 

## 5. Conclusions

In this study, we reviewed a method for assessing the seismic performance of waterproofing materials installed for the purpose of preventing water leakage in the underground pile walls of concrete building structures. Although the effect of preventing water leakage is advantageous when installed on the positive-side of underground structures, it has the disadvantage that it is very difficult to predict the leakage defect and determine the leakage pathing due to the geographical location where the waterproofing layer is installed. 

For this reason, it is necessary to verify the long-term durability of waterproofing materials applied to waterproofing walls. Among the long-term durability verification criteria, there are vibrations and behaviors that are continuously affected immediately after installation in concrete structures. Furthermore, there are seismic waves that occur intermittently and locally, but the scale of the transmitted wavelength can be large, causing a high degree of damage and even leading to collapse of the concrete structure. However, in the case of seismic waves, the collapse of the concrete structure is a matter beyond the limit of the performance that the waterproofing layer must respond to; thus, as a result of examining domestic and foreign related quality control standards to verify long-term durability against vibration, behavior, and earthquakes, it was confirmed that there is an absence of quality standards for waterproofing materials in this field. 

To develop criteria of evaluative performance for waterproofing materials with respect to seismic stress, we referred to a recently established national standard in Korea. In this reference, we conducted an analysis of the demand response spectrum of building structures required for the design of waterproof performance in response to vibration, behavior, and earthquake response, and of the demand response spectrum of the seismic design of non-structural elements. 

It was confirmed that the period or frequency of displacement, velocity, acceleration, etc. of the demand response spectrum of the model is transmitted in the form of aperiodic and highly complex wavelengths, and it was designed by reflecting these in the seismic performance. Based on these results, the demand response spectrum of the waterproofing layer, which is directly or indirectly attached to the concrete structure, was reinterpreted in terms of rotation, speed, and angle. By considering the analyzed rotation, speed, and angle, a new concept for a vibration and structural behavior testing method, consisting of a control box that can be driven and controlled, a 3-axis (X, Y, Z) seismic stress simulation device, a leakage checking device, and an experimental equipment device that can verify seismic response force, was implemented. 

A prototype of this equipment was produced based on the designed contents. As a first-stage preliminary experiment using a prototype of the manufactured test device, an evaluation was performed and demonstrated on siliceous powder-type coating waterproofing material for cement-based waterproofing (CM), urethane-based coating material (LM) for membrane waterproofing, and self-adhesive waterproofing sheet (SM) developed as a material for wall waterproofing. The demonstration experiment showed that the SM type had the highest anti-seismic response performance among the three material types, but more importantly, that the anti-seismic response properties of the respective types of waterproofing material were reflective of their inherent material properties. However, as this was intended to be a demonstration of the newly proposed testing method, the deformation/displacement of the materials were exaggerated beyond what is realistically expected during an actual seismic event in order to highlight that waterproofing materials of different properties respond differently to seismic load. The demonstration experiment served to indicate that the evaluation method is applicable for determining a waterproofing material’s seismic performance and response properties, but improvements in accuracy, the inclusion of different parameters considered for NSC, and future testing of more material types will be required.

Combining these results, it was possible to sufficiently confirm the need for a verification method for seismic resistance performance according to rotation, speed, and angle reinterpreted as vibration and behavior in order to secure the long-term durability required for waterproofing and earthquake response spectrum. It is judged that at this stage, additional research and review, such as setting a standard driving range for each material and establishing a standard test method for various waterproofing materials in the second stage, will be necessary based on the preliminary tests of the first stage with limited materials.

## Figures and Tables

**Figure 1 materials-14-05719-f001:**
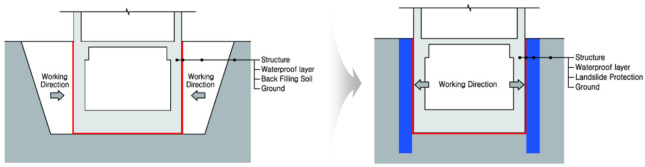
Conceptual diagram of single-side walls applied with existing waterproofing system technology.

**Figure 2 materials-14-05719-f002:**
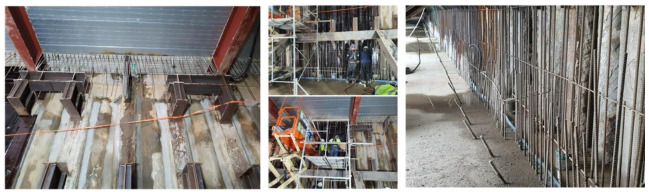
Examples of on-site waterproofing of pile walls (Dae-gu City, Korea).

**Figure 3 materials-14-05719-f003:**
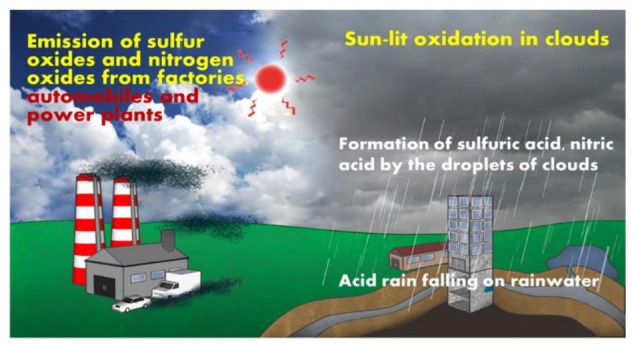
Groundwater pollution path.

**Figure 4 materials-14-05719-f004:**
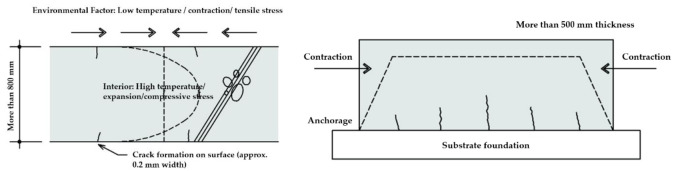
Effects on concrete substrate due to mechanical effects.

**Figure 5 materials-14-05719-f005:**
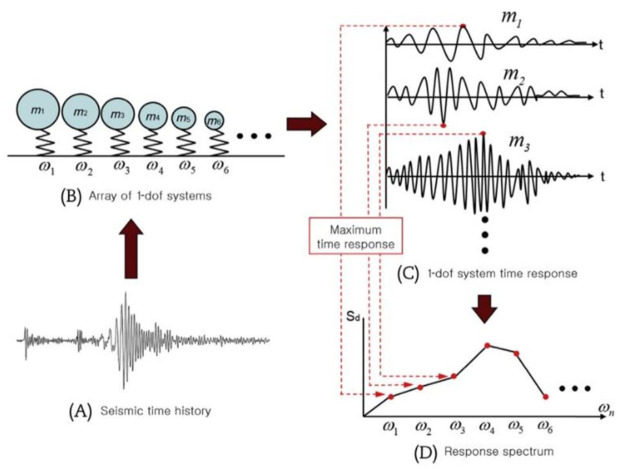
Basic concept of response spectrum (dof = degree of freedom).

**Figure 6 materials-14-05719-f006:**
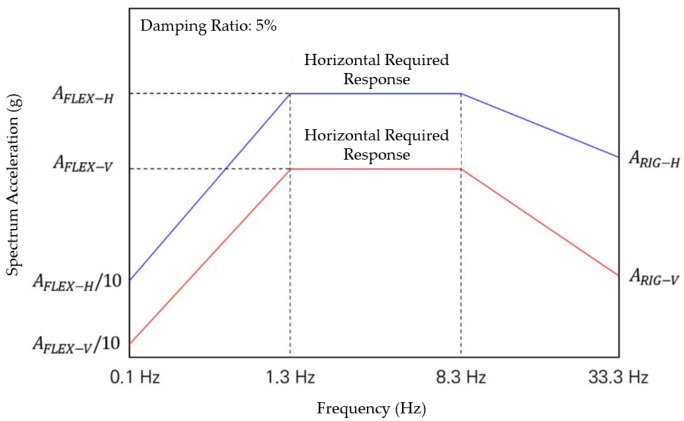
Required response spectrum for seismic table test.

**Figure 7 materials-14-05719-f007:**
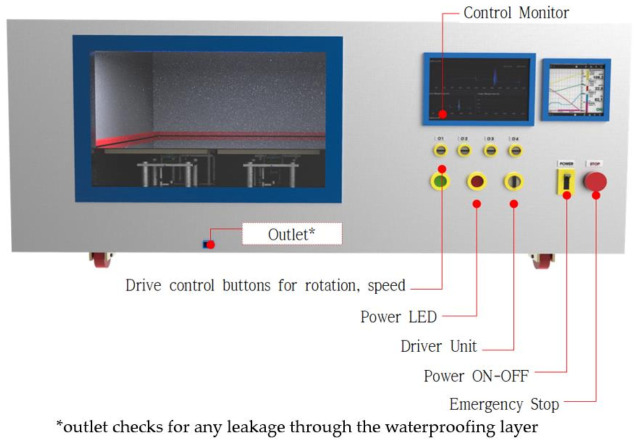
Experimental equipment design illustrated.

**Figure 8 materials-14-05719-f008:**
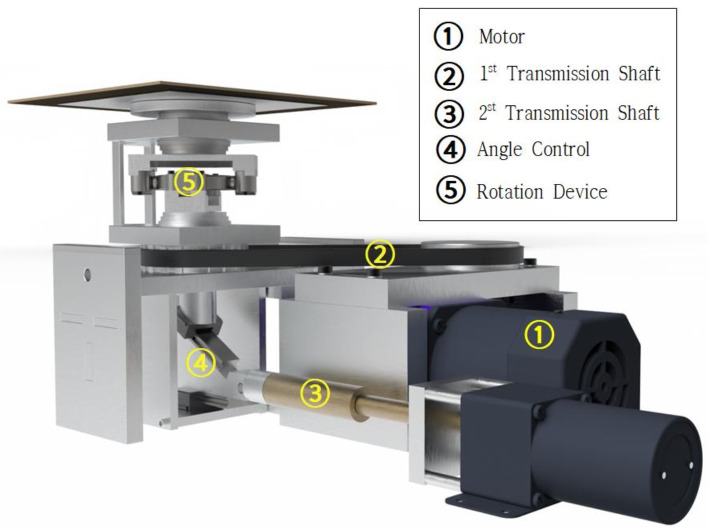
Device (simulator) concept.

**Figure 9 materials-14-05719-f009:**
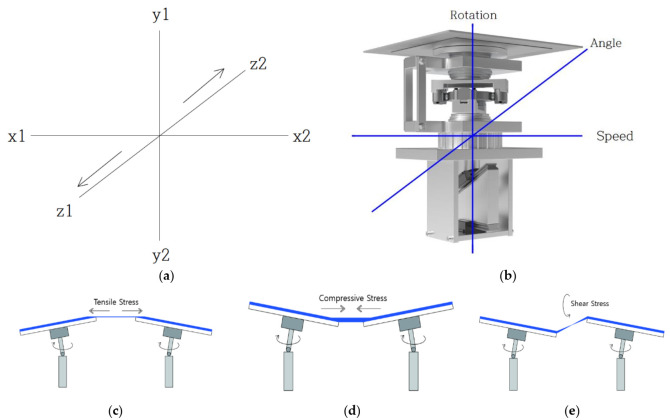
Concept of three-axis operation for stress generation based on rotation, speed, and angle using the motor device: (**a**) schematic diagram of triaxial stress generation, (**b**) schematic diagram of 3-axis drive of rotation, speed, and angle, (**c**) tensile stress simulation, (**d**) compressive stress simulation, (**e**) shear stress simulation.

**Figure 10 materials-14-05719-f010:**
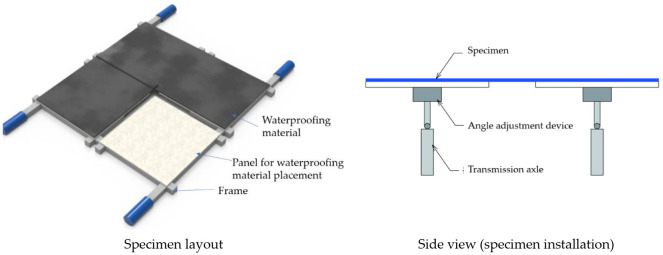
Schematic diagram of 3-axis driving stress verification of rotation, speed, and angle.

**Figure 11 materials-14-05719-f011:**
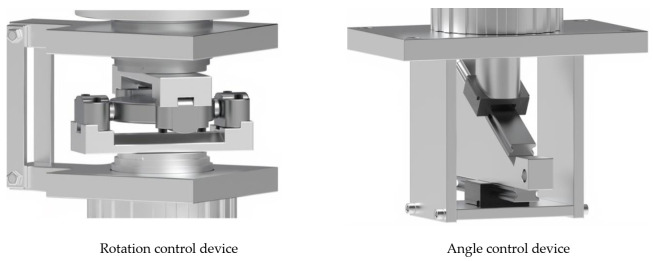
Rotation mechanism on the motor device illustrated.

**Figure 12 materials-14-05719-f012:**

Schematic diagram of possible layout settings of the rotation on the motor device.

**Figure 13 materials-14-05719-f013:**
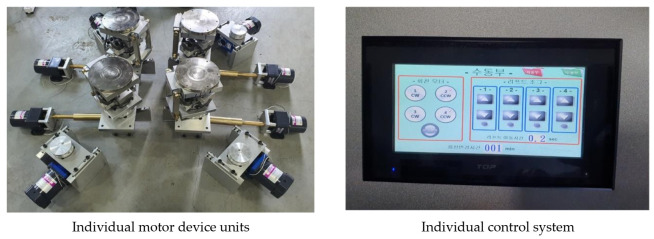
Control panel and the relevant motor devices.

**Figure 14 materials-14-05719-f014:**
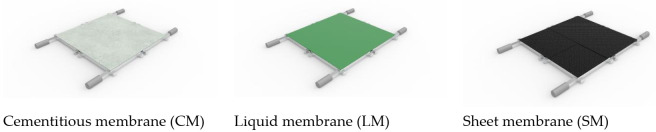
Specimens for testing.

**Figure 15 materials-14-05719-f015:**
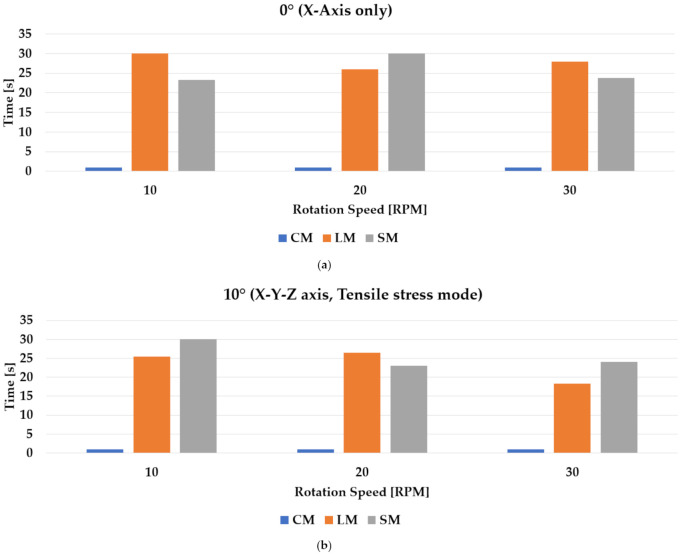
Specimens for testing, (**a**) 0° testing condition; (**b**) 10° testing condition, tensile stress mode; (**c**) 20° testing condition, shear stress mode.

**Table 1 materials-14-05719-t001:** Explanation of the four types of seismic waves [18].

Types	Pathing Type	Characteristic
P Wave	Concave	P-waves can run in solids and liquids. Out of P, S and L types of seismic waves, P-waves do the least harm to infrastructure, humans, nature, etc. The maximum speed of P-waves is around 14 km/s. Their average speed is 8 km/s.
S Wave	Concave	These waves can only run in solids, not liquids. Speed of S-waves range from 4 to 6 km/s. These waves move at moderate speed. After P-waves, S-waves plot on the seismograph. Paths of S-waves are concave. We know that the outer core of the Earth is made up of liquid, so these waves cannot penetrate the outer and inner core. Only S-Waves can cross the core.
L Wave	Convex	Speed of these waves is around 2 to 3 km/s. Path of surface waves is convex. These waves move at the slowest speed and cause the most damage. These waves remain on the surface of the Earth; that is why these waves can damage large buildings.
R Wave	Horizontal and vertical	Rayleigh waves move both vertically and horizontally on the surface of the Earth. These waves move in the vertical plane in the direction of motion. Rayleigh waves are rolling waves. These waves roll as water waves roll in sea or ocean; these are the most destructive waves. These seismic waves produce a long wave on seismographs.

**Table 2 materials-14-05719-t002:** Seismic wave types in response to seismic performance design of waterproofing materials [18].

Wave Types	Seismic Wave Illustrated
P Wave (Concave)	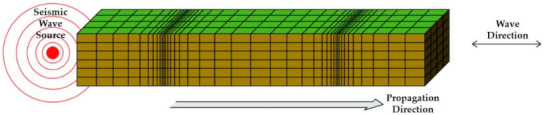
S Wave (Concave)	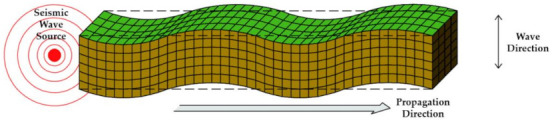
L Wave (Convex)	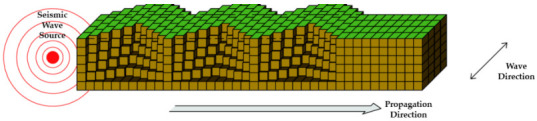
R Wave (Horizontal and vertical)	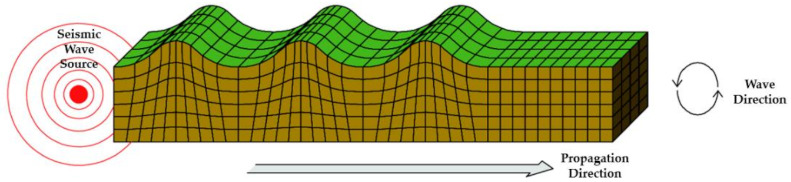

**Table 3 materials-14-05719-t003:** Summary of the experimental results (demonstration).

Angle	Rotation Speed (RPM)	Specimen No.	Average Result per Waterproofing Materials (4 Specimens)					
			CM		LM		SM	
			Result	Time	Result	Time	Result	Time
0° (X axis only)	10	1	O	30	O	30	O	30
		2	O	30	O	30	X	3
		3	X	9	O	30	O	30
		4	O	30	O	30	O	30
		Average	O	24.75	O	30	O	23.25
	20	1	O	30	O	30	O	30
		2	X	8	O	30	O	30
		3	X	9	O	30	O	30
		4	O	30	O	30	O	30
		Average	X	19.25	O	30	O	30
	30	1	X	6	O	30	O	30
		2	X	8	O	30	X	5
		3	X	9	O	30	O	30
		4	X	1	O	30	O	30
		Average	X	6	O	30	O	23.75
10° (X-Y-Z axis, Tensile Stress Mode)	10	1	X	1	O	30	O	30
		2	X	1	O	30	O	30
		3	X	1	X	23	O	30
		4	X	1	X	19	O	30
		Average	X	1	X	25.5	O	30
	20	1	X	1	X	21	O	30
		2	X	1	O	30	O	30
		3	X	1	X	25	O	30
		4	X	1	O	30	X	2
		Average	X	1	X	26.5	O	23
	30	1	X	1	X	21	O	30
		2	X	1	X	14	O	30
		3	X	1	X	16	O	30
		4	X	1	X	22	X	6
		Average	X	1	X	18.25	O	24
20° (X-Y-Z axis, Shear Stress Mode)	10	1	X	1	X	15	X	30
		2	X	1	X	12	X	1
		3	X	1	X	16	X	5
		4	X	1	X	13	X	30
		Average	X	1	X	14	X	16.5
	20	1	X	1	X	14	O	30
		2	X	1	X	18	X	5
		3	X	1	X	16	X	3
		4	X	1	O	30	X	4
		Average	X	1	X	19.5	X	10.5
	30	1	X	1	X	13	O	3
		2	X	1	X	14	X	4
		3	X	1	X	19	X	3
		4	X	1	X	12	O	5
		Average	X	1	X	14.5	X	3.75

## Data Availability

Not applicable.

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
