# Peer review of "Anti-Seismic Performance Evaluation of Waterproofing Materials for Underground Pile Wall Structures"

_materials, 2021, doi:10.3390/ma14195719_

Round 1

Reviewer 1 Report

Substantive comments:

What were the test conditions (humidity, temperature)? Under real conditions, at low temperatures, some materials (e.g. bituminous coatings) may be much more brittle than at room temperature. Please complete this information and comment on it.

Were the amounts of deformation/displacement in the material tests not too high in relation to the actual seismic conditions? Please comment on it.

Do the authors project checking the results in practical conditions (which may not be possible) or in numerical simulation? Please comment on it.

Editorial comments:

Please provide the source of the photos in Figure 2.

Line 153: it should be Richter.

Table 1: it is Km/sec, it should be km/s; there is S-Waves, there should be S-waves; there should be no spaces in km/s.

In Table 2, the first column is wrong formatted and the descriptions on the right side are incomplete.

In paragraphs 227-243, the references should be made to the number of the reference, not the titles and authors.

Heading p. 3.5.1. need to be moved to the next page. The same concerns for heading 4.

In Fig. 9 the descriptions need to be formatted. The same concerns Fig. 14.

In Fig. 15 the vertical axes should be described (Time [s]).

The points should be correctly described (or deleted): Institutional Review Board Statement, Informed Consent Statement, Acknowledgments.

Abbreviations are repeated 2 times.

Author Response

The authors of article materials–1382626 would like to extend their sincerest gratitude to the reviewers for taking time out of their busy schedule to review our paper. Much thanks to your efforts, comments and contribution, we hope that the paper has improved in quality and clarity. The below lists the comments provided by the reviewers and the respective response and details as to how the comments were address and applied in the revised article.

Substantive comments:

Comment 1

What were the test conditions (humidity, temperature)? Under real conditions, at low temperatures, some materials (e.g. bituminous coatings) may be much more brittle than at room temperature. Please complete this information and comment on it.

Response 1

For this point, details on the ambient conditions for testing has been included in section 4.4.2, lines 440 to 443. As the reviewer has aptly pointed out, certain material types (such as bituminous coatings) are indeed very sensitive to temperature changes and are subject to property changes, and it was deemed important to stress in the article that the testing was conducted in standard room temperature setting to ensure that materials will not undergo these changes before and during testing.

Comment 2

Were the amounts of deformation/displacement in the material tests not too high in relation to the actual seismic conditions? Please comment on it.

Response 2

As far as actual deformation is concerned for the specimens during seismic load simulation during the testing, there is actually a minimal movement between the panels, and the motors are mainly applying only vibration to the panels after they have been set according to their respective angles. As far as the angles are concerned, it is as the reviewer mentions that for normal constructions, waterproofing materials would not be suddenly twisted into a 20° angle under normal circumstances (if the structure was being destroyed, then such conditions may be applicable), but the main focus for this particular study is for application to pile walls, where due to the nature the secant piles, the surface is not smooth and waterproofing materials have to be applied on various types of surfaces, ones that are angled with corners and overlaps. In such cases, waterproofing sheets (for example) that are commonly known to be able to secure high and stable adhesion on regular surfaces may be susceptible to adhesion failure due to the initial lack of proper installation from just minor vibration effects (such as earthquakes) as it commonly is difficult to secure proper workmanship on vertical angular column type walls. Furthermore, the conditions applied in this testing is purely for demonstration purposes with the intent to clearly highlight the difference of seismic performance of waterproofing materials based on their properties, and was exaggerated beyond what is realistically expected during actual earth quakes (on this note, further explanation has been included in Lines 539~543 of the revised manuscript), and when conducting objective testing for performance comparison between different products, more realistic conditions will be used.

Comment 3

Do the authors project checking the results in practical conditions (which may not be possible) or in numerical simulation? Please comment on it.

Response 3

The article is part of an ongoing government research project being conducted by the by Korea Land Transport Technology Institution and the authors plan on continue reporting the results of this study through future papers. Finite element analysis (numerical simulation) is a part of the study and the results of the comparative analysis will be provided in future papers, but for the time being, the authors feel that it is important to first introduce the concept, experimental methodology and testing apparatus through this as numerical analysis on top of the demonstration procedure may overwhelm the interested readers.

Editorial comments:

Comment 1

Please provide the source of the photos in Figure 2.

Response 1

Source of the photos have been included in the Figure 2 title, but details of the location cannot be disclosed as the work is being conducted by a private waterproofing company.

Comment 2

Line 153: it should be Richter.

Response 2

Term has been revised accordingly

Comment 3

Table 1: it is Km/sec, it should be km/s; there is S-Waves, there should be S-waves; there should be no spaces in km/s.

Response 3

Table 1 has been revised accordingly

Comment 4

In Table 2, the first column is wrong formatted and the descriptions on the right side are incomplete.

Response 4

Table 2 column has been revised. The authors believe that in the process of editing by the editorial division of MDPI, the table format was changed.

Comment 5

In paragraphs 227-243, the references should be made to the number of the reference, not the titles and authors.

Response 5

Reference format has been revised for lines 227~243

Comment 6

Heading p. 3.5.1. need to be moved to the next page. The same concerns for heading 4.

Revised

Comment 7

In Fig. 9 the descriptions need to be formatted. The same concerns Fig. 14.

Comment 8

In Fig. 15 the vertical axes should be described (Time [s]).

Response 8

The respective Axis have been properly labelled for Figure 15.

Comment 9

The points should be correctly described (or deleted): Institutional Review Board Statement, Informed Consent Statement, Acknowledgments.

Response 9

The above mentioned points have been removed.

Comment 10

Abbreviations are repeated 2 times.

Response 10

Repeated abbreviation has been removed

The authors would like to express their thanks once again for the reviewer’s time and valuable feed back on this manuscript.

Reviewer 2 Report

In the Reviewer opinion the research paper entitled “Anti-seismic performance evaluation of waterproofing materials for positive-side wall and pile wall of underground concrete structures” is good.

This study introduces and demonstrates the application of an experimental regime for anti-seismic performance evaluation of waterproofing materials to used for concrete pile walls. Concrete pile walls are subject to high degree of seismic load, and the occurring stress can affect the waterproofing integrity of the structure, but there is currently no existing methodology or standard for evaluating this property of waterproofing materials. To propose and conduct this evaluation, a new testing apparatus was designed and manufactured intended to be able to test an installed waterproofing material’s seismic resistance performance.

Some comments which greatly enhance the understanding of the paper and its value are presented below. Specific issues that require further consideration are:

  1. The title of the manuscript is matched to its content but it is too long.
  2. The Introduction generally covers the cases.
  3. The methodology was clearly presented.
  4. In the Reviewer’s opinion, the current state of knowledge relating to the manuscript topic has been presented, but the author's contribution and novelty are not enough emphasized.
  5. Experimental program and results looks interesting and was clearly presented.
  6. In the Reviewer’s opinion, the bibliography, comprising 18 references, is rather not representative.
  7. An analysis of the manuscript content and the References shows that the manuscript under review constitutes a summary of the Author(s) achievements in the field.
  8. In the Reviewer’s opinion the manuscript is well written, and it should be published in the journal after major revision.

Author Response

The authors of article materials–1382626 would like to extend their sincerest gratitude to the reviewers for taking time out of their busy schedule to review our paper. Much thanks to your efforts, comments and contribution, we hope that the paper has improved in quality and clarity. The below lists the comments provided by the reviewers and the respective response and details as to how the comments were address and applied in the revised article.

  1. The title of the manuscript is matched to its content but it is too long.

Response: the title of the manuscript was revised to read as the following; “Anti-seismic performance evaluation of waterproofing materials for underground pile wall structures”

  1. The Introduction generally covers the cases.

Response: Thank you for the comment

  1. The methodology was clearly presented.

Response: Thank you for the comment

  1. In the Reviewer’s opinion, the current state of knowledge relating to the manuscript topic has been presented, but the author's contribution and novelty are not enough emphasized.

Response: The paper intended novelty and contribution of this research has been more clearly stated in the revised manuscript, lines 59~72

  1. Experimental program and results looks interesting and was clearly presented.

Response: Thank you for the comment

  1. In the Reviewer’s opinion, the bibliography, comprising 18 references, is rather not representative.

Response: more references have been added

  1. An analysis of the manuscript content and the References shows that the manuscript under review constitutes a summary of the Author(s) achievements in the field.

Response: the article does not provide a separate section within the manuscript that allows the promotion or summarization of the Author(s) achievements or credentials aside from personal profile sections of the MDPI website or researchgate, where the authors will glad comply with the request.  

  1. In the Reviewer’s opinion the manuscript is well written, and it should be published in the journal after major revision.

Response: Thank you for the comment

The authors would like to express their thanks once again for the reviewer’s time and valuable feed back on this manuscript.

Round 2

Reviewer 2 Report

Authors corrected manuscript follow to my suggestion. In my opinion article should be published in the Journal